# Dynamic–Static Representation Learning with Mamba-Enhanced Diffusion for Temporal Knowledge Graph Reasoning

## Abstract

Temporal Knowledge Graph (TKG) reasoning aims to predict future missing facts based on historical evidence. Prior studies on graph learning and logical rules often overlook global latent semantics and struggle with long-range dependencies, particularly under sparse or unseen facts. To address these limitations, we propose DSEE-MDiff, which frames TKG reasoning as selecting informative history and denoising future signals. Specifically, a Dynamic–Static Entity Selection encoder captures global semantic evolution alongside local structural cues, while a Mamba-based diffusion module injects and removes noise with a selective state-space model to better recover long-range dependencies and mitigate sparsity. The two outputs are fused for prediction through a ConvTransE decoder. Experiments on four public datasets demonstrate that DSEE-MDiff achieves state-of-the-art performance across multiple metrics, validating the effectiveness of the proposed approach.

## Availability of Code and Data

The implementation of DSEE-MDiff and all experimental datasets are publicly available at: `https://anonymous.4open.science/r/DSEE-MDiff-E8B1`.

## 1 Introduction

Knowledge graphs (KGs) represent real-world facts in a structured form. A static KG stores facts as triples $(s, r, o)$, where $s$ is the head entity, $r$ is the relation, and $o$ is the tail entity. To capture the temporal dynamics of facts, temporal knowledge graphs (TKGs) extend this representation to quadruples $(s, r, o, t)$, where $t$ denotes the timestamp. For example, the quadruple (Trump, *claim*, *end war*, 2025-08-19) indicates that Trump made the claim on August 19, 2025. Facts with the same timestamp form a snapshot, and the temporal ordering of snapshots composes a TKG. TKGs serve as the foundation for many NLP tasks, and their quality directly influences downstream applications such as question answering (Saxena et al., 2020; Liu et al., 2023b), information retrieval (Zamiri et al., 2024), and recommendation (Wang et al., 2024b).

Similar to static KGs, TKGs are inherently incomplete. Temporal knowledge graph reasoning (TKGR) aims to predict missing entities or relations under temporal conditions, thereby alleviating incompleteness. TKGR is commonly studied under two settings: *interpolation*, which infers missing facts within the observed time span, and *extrapolation*, which predicts facts beyond the observed horizon. Figure 1 illustrates an extrapolation example. Because extrapolation better reflects real-world applications and offers broader potential, we focus on extrapolation in this work.

Accurate prediction of future facts requires effectively leveraging historical snapshots. Existing methods attempt to mine temporal patterns from history. For instance, RE-GCN (Li et al., 2021) models temporal dependencies for entity–relation evolution; RE-NET (Jin et al., 2020) encodes histories centered on the target entities; xERTE (Han et al., 2020b) samples time-aware neighborhoods to construct explanatory subgraphs; and TLogic (Liu et al., 2022) extracts temporal logic rules via timed random walks. These approaches advance TKGR by encoding event sequences into entity

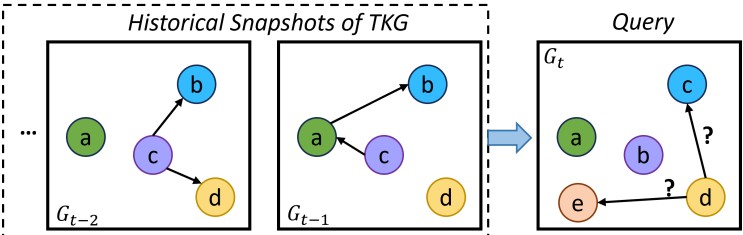

Figure 1: An example of temporal reasoning. At time $t$, the task is to predict whether entity $d$ will interact with $c$ or $e$. Because $d$ has no prior interaction with $a$, the prediction is more challenging.

representations and exploiting local snapshot structures. However, they still face challenges in capturing *global* latent semantics. First, subgraphs from different snapshots are often modeled independently, overlooking cross-snapshot interactions and semantic evolution over time. Second, most models lack adaptive mechanisms to prioritize semantic signals at different levels. Jointly modeling and integrating graph-structural features with global semantics therefore remains a key challenge for TKGR.

In practice, many facts appear with low frequency or may even be unseen during testing. Recent studies employ diffusion techniques, such as noise injection and denoising, to enhance generalization on sparse or unseen facts (Cai et al., 2024; Cao et al., 2025). However, when temporal conditioning is weak or only implicit, these methods often fail to fully exploit semantic dependencies among facts, such as concurrency or precedence. As the history window expands, denoising tends to rely primarily on local signals, causing long-range interactions to diminish and ultimately limiting extrapolation performance. Although some designs incorporate frequency priors, entity-level cues, or snapshot-level features, they still struggle to capture latent semantic connections across facts.

To overcome these limitations, we propose DSEE-MDiff, an encoder–decoder framework for temporal knowledge graph completion with three main components: (i) a dynamic–static entity selection encoder, (ii) a Mamba-driven diffusion module, and (iii) a decoder. In the encoding stage, we treat global and local snapshots differently: the dynamic encoder captures semantic evolution across history, while the static encoder emphasizes graph-structural semantics such as topology and relational constraints within subgraphs. A learnable selection mechanism adaptively fuses these two signals. We then introduce Mamba-Diffusion to improve extrapolation on unseen facts. By gradually injecting and removing noise in the embedding space, it synthesizes plausible new facts, while the selective state-space mechanism of Mamba preserves long-range semantic dependencies during denoising, thereby mitigating information decay in long histories. Finally, a ConvTransE decoder scores candidate entities and generates predictions.

Our contributions can be summarized as follows:

- We present DSEE-MDiff, a temporal reasoning framework that couples a dynamic–static entity selection encoder with a Mamba-based diffusion module. This design enables the model to capture both graph structure and sequence-level semantics, thereby improving prediction on rare and unseen facts.

- We design a dynamic–static selection encoder that integrates complementary signals from global history and local subgraphs, adaptively emphasizing the portions of history most relevant to the query.

- We introduce a Mamba-Diffusion denoiser that combines Mamba with lightweight Transformer layers, enhancing the ability to model long-range dependencies during the denoising process.

- Extensive experiments on four public datasets demonstrate consistent gains over strong baselines, with DSEE-MDiff achieving state-of-the-art performance in multiple settings.

## 2 RELATED WORK

### 2.1 TEMPORAL KNOWLEDGE GRAPH REASONING

Extrapolative reasoning on TKGs aims to predict future events from historical evidence and structural patterns in temporal snapshots. Early studies relied on simple statistical signals. For example, CyGNet (Zhu et al., 2021) introduces a time-aware copy mechanism to capture repetitive one-hop patterns, and RE-NET (Jin et al., 2020) employs recurrent networks with neighborhood aggregation to encode local history.

As research progressed, it became evident that evolutionary patterns are crucial for extrapolation, as they reflect dynamic trends in historical events. RE-GCN (Li et al., 2021) and its extension CEN (Li et al., 2022b) provide unified modeling of graph sequences to improve short-term temporal dependency learning. To jointly capture short- and long-term history, TiRGN (Li et al., 2022a) and HiSMatch (Li et al., 2022c) adopt dual encoders for temporal dependencies and periodic signals, combined with local neighbor aggregation and query-conditioned global statistics. EvoExplore (Zhang et al., 2022) explicitly builds historical context to model complex event evolution; TLogic (Han et al., 2020b) mines temporal logical rules via timed random walks; and TECHS (Lin et al., 2023) integrates temporal encodings with heterogeneous attention and continual message passing to jointly embed structural and temporal dynamics.

Recent advances also emphasize noise filtering and the fusion of multi-scale temporal patterns. Re-Temp (Wang et al., 2023) introduces a query-driven pruning mechanism to retain the most relevant historical signals, while LSEN (Wang et al., 2024a) jointly models short- and long-range history. CognTKE (Chen et al., 2025), inspired by dual-process theory, employs an attention-based TCR-Digraph to fuse local paths with global embeddings and enable zero-shot reasoning. LogCL (Chen et al., 2024) improves robustness under noisy histories through local and global contrastive objectives, whereas TRCL (Liu et al., 2025) unifies periodic and aperiodic signals with a recursive convolutional encoder and a global history matrix, further mitigating noise with contrastive learning.

Diffusion-based approaches have also been investigated. DiffCLR (Liu et al., 2024) applies multi-step diffusion to inject uncertainty and form distributions that better capture multi-faceted query semantics. DiffuTGK (Cai et al., 2024) reframes TKG reasoning as sequence prediction: it conditions on historical events, perturbs target facts with Gaussian noise in the forward process, and restores them through reverse-time denoising.

### 2.2 MAMBA MODELS ON DISCRETE DATA

State space models (SSMs), inspired by continuous-time dynamical systems (Gu et al., 2021a;b), combine the parallel training efficiency of CNNs with the fast inference of RNNs. Building on SSMs, Mamba (Gu & Dao, 2023) introduces parameterized state matrices and hardware-aware parallelism, achieving strong performance on language modeling, DNA sequences, and audio signals. Following this line of work, a family of Mamba variants has emerged, extending applications to computer vision (Zhu et al., 2024) and time-series modeling. Owing to its ability to support content-based reasoning while maintaining efficiency, Mamba has been recognized as a promising alternative to Transformers for long-sequence processing. Consequently, researchers have adapted Mamba to a variety of NLP tasks, including question answering (Lieber et al., 2024; He et al., 2024) and summarization (Bronnec et al., 2024; Ren et al., 2024). DenseMamba (He et al., 2024) selectively routes shallow hidden states to deeper layers to enrich information flow, while SAMBA (Ren et al., 2024) integrates Mamba with sliding-window attention, enabling selective compression into recurrent states while preserving precise recall.

Despite these strengths, pure Mamba encoders can underperform Transformer architectures in tasks that require non-standard retrieval or in-context learning (Park et al., 2024). This limitation has motivated hybrid designs that combine Transformer blocks with Mamba modules to exploit their complementary advantages for long-range dependency modeling. Nevertheless, applications of Mamba to temporal knowledge graph reasoning remain largely unexplored.

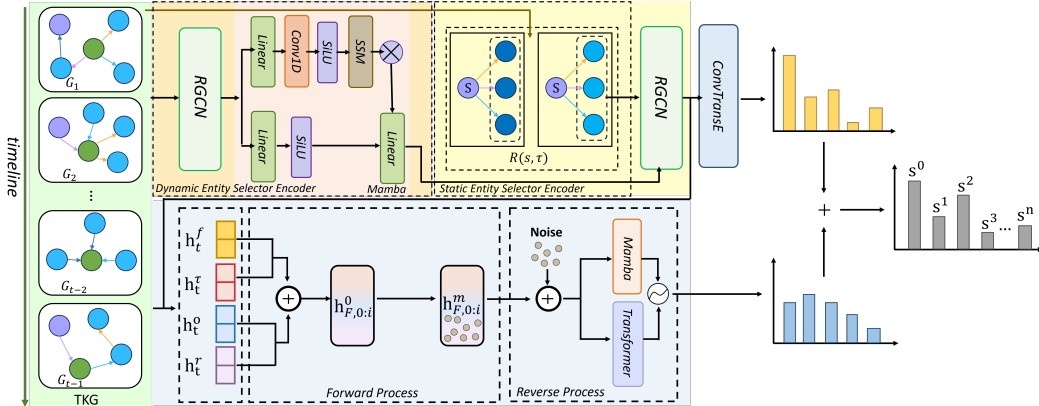

Figure 2: Overall framework of the proposed DSEE-MDiff for TKG extrapolation. The Dynamic–Static Entity Selection Encoder produces entity/relation representations; the Mamba-driven diffusion denoises fact-level sequences; and the ConvTransE performs decoding. Both are trained jointly and their scores are combined at inference.

## 3 METHOD

Our framework adopts an encoder–decoder paradigm consisting of three main components (Fig. 2): (i) a Dynamic–Static Entity Selection encoder that integrates global semantic evolution with local structural information from historical snapshots; (ii) a Mamba-driven diffusion module that perturbs fact-level representations through noise injection and denoising to enhance generalization on sparse or unseen facts; and (iii) a ConvTransE decoder that assigns scores to candidate entities. The diffusion module and the ConvTransE decoder are trained jointly, and their outputs are aggregated during inference to produce the final prediction.

### 3.1 PRELIMINARIES

The goal of temporal knowledge graph (TKG) reasoning is to predict future facts based on historical observations. A TKG is formally defined as $\mathcal{G} = (\mathcal{E}, \mathcal{R}, \mathcal{T}, \mathcal{F})$, where $\mathcal{E}$, $\mathcal{R}$, $\mathcal{T}$, and $\mathcal{F}$ denote the sets of entities, relations, timestamps, and facts, respectively. Equivalently, $\mathcal{G}$ can be regarded as an ordered sequence of snapshots $\{G_0, \ldots, G_T\}$, where each snapshot $G_t \subseteq \mathcal{E} \times \mathcal{R} \times \mathcal{E}$ contains all quadruples $(s, r, o, t)$ valid at time $t$. Following common practice (Jin et al., 2020), for each fact $(s, r, o, t)$ we also include its inverse $(o, r^{-1}, s, t)$ to preserve bidirectional semantics.

Given a future query $q = (s, r, ?, t_q) \in \mathcal{Q}_{t_q}$ (or symmetrically $q = (?, r, o, t_q)$), the task is to estimate the probability of the missing entity conditioned on historical snapshots:

$$p\big(o \mid s, r, t_q, G_{0:t_q-1}\big), \qquad G_{0:t_q-1} = \{G_0, \ldots, G_{t_q-1}\}. \tag{1}$$

Here $d$ denotes the embedding dimension of entities and relations.

## 4 DYNAMIC–STATIC ENTITY SELECTION ENCODER

As discussed in the introduction, the adjacent $l$ historical snapshots $\{G_{t_q-l}, \ldots, G_{t_q-1}\}$ are related to the query $(s_q, r_q, ?, t_q)$. However, different snapshots contribute unequally to the query. To effectively capture entity evolution over time and to distinguish the contribution of each snapshot, we design a Dynamic–Static Entity Selection encoder that adaptively mines query-relevant historical information. The dynamic branch extracts evolutionary representations of entities from a global perspective, and through a selection mechanism, highlights salient features. The static branch complements this by emphasizing associations within local temporal snapshots.

**Dynamic Entity Selection Encoder** At each historical snapshot, our goal is to retain as many query-relevant facts as possible. To this end, we first apply RGCN Schlichtkrull et al. (2018) to aggregate

multi-relational and multi-hop neighborhood information, yielding structured entity representations that serve as graph-structural features for subsequent dynamic selection:

$$\mathbf{h}_s^{(l+1)} = \text{RGCN}\big(\mathbf{h}_s^{(l)}, \mathbf{h}_r^{(l)}, \mathbf{h}_o^{(l)}\big), \tag{2}$$

where $\mathbf{h}_s^{(l)} \in \mathbb{R}^{|\mathcal{E}| \times d}$ and $\mathbf{h}_r^{(l)} \in \mathbb{R}^{2|\mathcal{R}| \times d}$ denote entity and relation representations at layer $l$. Let the output of RGCN be $\mathbf{h}_s'$. We then feed $\mathbf{h}_s'$ into Mamba to dynamically adjust entity importance and further model intra-snapshot dependencies. Leveraging selective state-space updates, Mamba focuses on key features rather than treating all historical information uniformly. Compared with RGCN alone, this allows better modeling of temporal evolution, enabling efficient temporal dependency learning. The computation is

$$\mathbf{h}_s' = \text{SSM}\big(\text{SiLU}(\text{Conv}(\text{MLP}(\mathbf{h}_s)))\big), \tag{3}$$

where $\text{SiLU}(\cdot)$ is the activation function, and $\mathbf{h}_s' \in \mathbb{R}^{|\mathcal{E}| \times d}$ is the dynamic entity representation.

**Static Entity Selection Encoder**  The static entity selection encoder complements the dynamic branch by incorporating relation statistics that may not be fully captured by local structural features. For snapshots $\{G_{t_q-l}, \ldots, G_{t_q-1}\}$, we compute static entity representations via weighted aggregation across time. Specifically, at timestamp $\tau$, for each entity $s$, we mean-pool its incident relations to obtain a reference vector $\overline{\mathbf{h}}_r$, and combine it with entity embeddings from the previous $l$ timestamps. A feed-forward network produces weights $\mathbf{h}_j^d$ to guide the aggregation:

$$\mathbf{h}_s^t = \mathbf{h}_0^d \mathbf{h}_s'^t + \sum_{i=1}^{l} \mathbf{h}_i^d \mathbf{h}_s'^{t-i}, \tag{4}$$

$$\mathbf{h}_j^d = \text{softmax}\big(\text{MLP}\big(\mathbf{h}_s'^{t-j} + \overline{\mathbf{h}}_r\big)\big), \quad j \in [0, l], \tag{5}$$

$$\overline{\mathbf{h}}_r = \frac{1}{|\mathcal{R}(s, \tau)|} \sum_{r \in \mathcal{R}(s, \tau)} \mathbf{h}_r, \tag{6}$$

where $\mathcal{R}(s, \tau)$ denotes the set of relations incident to entity $s$ at time $\tau$. The resulting entity embedding matrix is denoted by $\mathbf{H}_s^t$, and the relation embedding matrix by $\mathbf{H}_r^t$.

In summary, the dynamic encoder autonomously highlights informative entity signals from historical snapshots, while the static encoder provides global constraints through relation statistics. Their integration leverages both semantic evolution and structural associations, yielding more comprehensive entity representations.

### 4.1 Diffusion with Mamba

The entity and relation representations produced by the Dynamic–Static Entity Selection encoder are fed into a diffusion module. This module injects noise in a forward process and removes it in a reverse process, thereby improving generalization on sparse or unseen facts. We treat entities, relations, timestamps, and frequency signals as atomic units for noising and denoising, enabling the model to capture the dynamic evolution of temporal knowledge graphs more comprehensively. Inspired by fact-level noising strategies Long et al. (2024), we extend them by incorporating temporal information and occurrence frequency. The Mamba–Diffusion module consists of two stages: forward diffusion and reverse denoising.

**Forward Diffusion Process**  The forward process gradually injects noise into fact representations. We use the most recent $n$ relations and objects associated with the query subject $s$, denoted as $Q_{0:n} = \{Q_{0:n-1}, Q_t\}$, where $Q_t = (s, r_t, o_t, t_t)$ is the current query. For each historical fact $(s, r_i, o_i, t_i)$, the embeddings of $o_i$ and $r_i$ are drawn from $\mathbf{H}_s^t$ and $\mathbf{H}_r^t$. To encode temporal evolution, we introduce a time-difference embedding:

$$\mathbf{h}_\tau^t = \text{Emb}_\tau(t_q - t), \tag{7}$$

where $t_q$ is the query timestamp and $t$ is the historical timestamp. Frequency information is encoded similarly:

$$\mathbf{h}_f^t = \text{Emb}_f(f_q - f), \tag{8}$$

where $f_q$ is the maximum frequency in the sequence and $f$ is the frequency of the current fact. We concatenate relation, object, time-difference, and frequency-difference features and pass them through a feed-forward network:

$$\mathbf{h}_F = \mathrm{MLP}\big(\mathbf{h}_r^t \oplus \mathbf{h}_o^t \oplus \mathbf{h}_\tau^t \oplus \mathbf{h}_f^t\big). \tag{9}$$

Gaussian noise is then added for $m$ steps up to a maximum $M$:

$$q\big(\mathbf{h}_{F,i}^m \mid \mathbf{h}_{F,i}^0\big) = \begin{cases} \mathbf{h}_{F,i}^0, & i < n, \\ \sqrt{\bar{\lambda}_m}\,\mathbf{h}_{F,i}^0 + \sqrt{1 - \bar{\lambda}_m}\,\epsilon, & i = n, \end{cases} \tag{10}$$

$$1 - \bar{\lambda}_m = \delta \cdot \Big(\lambda_{\min} + \frac{m-1}{M-1}(\lambda_{\max} - \lambda_{\min})\Big), \tag{11}$$

where $\epsilon \sim \mathcal{N}(0,1)$ and $\delta, \lambda_{\min}, \lambda_{\max}$ are hyperparameters.

**Reverse Denoising Process** In the reverse process, the diffusion head progressively removes noise. To balance global and local cues, we fuse outputs from Transformer and Mamba paths using a learnable gate. Mamba performs selective state updates to filter irrelevant noise and retain salient features during denoising. The conditional denoising distribution is:

$$p_\theta\big(\hat{\mathbf{h}}_F^{m-1} \mid \hat{\mathbf{h}}_F^m, \mathbf{h}_o^t, \mathbf{h}_r^t, \mathbf{h}_\tau^t\big) = \mathcal{N}\big(\hat{\mathbf{h}}_F^{m-1}; \mu_\theta(*), \Sigma_\theta(*)\big), \tag{12}$$

where $* = \{\hat{\mathbf{h}}_F^m, m, \mathbf{h}_o^t, \mathbf{h}_r^t, \mathbf{h}_\tau^t\}$. The mean is a gated combination of Transformer and Mamba encoders:

$$f_{\theta 1}(*) = \mathrm{Transformer}\big(\bar{\mathbf{h}}_F^m\big), \qquad f_{\theta 2}(*) = \mathrm{Mamba}\big(\bar{\mathbf{h}}_F^m\big), \tag{13}$$

$$f_\theta(*) = \Theta\,f_{\theta 1}(*) + (1 - \Theta)\,f_{\theta 2}(*), \tag{14}$$

with $\bar{\mathbf{h}}_F^m = \hat{\mathbf{h}}_F^m \oplus \mathbf{h}_r^t \oplus \mathbf{h}_\tau^t \oplus \mathrm{Emb}(m)$, and $\mathrm{Emb}(m)$ denoting a learnable step embedding. Mamba captures global dependencies and enhances robustness to noise, complementing the local modeling of the Transformer.

Finally, $f_\theta$ can be trained to directly predict the clean representation $\hat{\mathbf{h}}_F^0$ from a noisy state $\bar{\mathbf{h}}_F^m$, allowing inference from the final step:

$$\hat{\mathbf{h}}_F^0 = f_\theta(\bar{\mathbf{h}}_F^M, M). \tag{15}$$

## 4.2 DECODING

The final prediction is obtained by combining the outputs of ConvTransE and Mamba–Diffusion. During training, we optimize the two components jointly.

We employ ConvTransE Shang et al. (2019) as the base decoder. Specifically, ConvTransE applies a 1D convolution to the concatenation of the subject and relation embeddings, followed by scoring all candidate entities:

$$S_{\mathrm{ct}} = \mathrm{softmax}\big(\mathrm{ConvTransE}(\mathbf{h}_s^t, \mathbf{h}_r^t) \cdot (\mathbf{H}_s^t)^\top\big), \tag{16}$$

$$\mathcal{L}_{\mathrm{ct}} = -\sum_{i=0}^{Q_t} g_i \log S_{\mathrm{ct}}, \tag{17}$$

where $Q_t$ is the number of queries at timestamp $t$, $g_i \in \mathbb{R}^{|\mathcal{E}|}$ is the one-hot label vector for entities, $\top$ denotes transpose, and $\mathbf{H}_s^t \in \mathbb{R}^{|\mathcal{E}| \times d}$ is the entity embedding matrix.

For the Mamba–Diffusion module, we define its loss as:

$$S_{\mathrm{diff}} = \mathrm{softmax}\big(f_\theta(\bar{\mathbf{h}}_F, m)_n \cdot (\mathbf{H}_s^t)^\top\big), \tag{18}$$

$$\mathcal{L}_{\mathrm{diff}} = -\sum_{i=0}^{Q_t} g_i \log S_{\mathrm{diff}}, \tag{19}$$

where $f_\theta(\bar{\mathbf{h}}_F, m)_n \in \mathbb{R}^{1 \times h}$ is the representation of the target object at step $m$.

The two outputs are aggregated to form the final prediction:

$$\mathbf{P}(o \mid s, r, \tau) = S_{\mathrm{ct}} + S_{\mathrm{diff}},$$
$$o_\tau = \mathrm{argmax}\,\mathbf{P}(o \mid s, r, \tau). \tag{20}$$

The overall training objective jointly optimizes both losses:

$$\mathcal{L} = \mathcal{L}_{\mathrm{ct}} + \mathcal{L}_{\mathrm{diff}}. \tag{21}$$

## 5 EXPERIMENTS

Table 1: Details of the four TKG datasets.

| Datasets | #Entities | #Relations | #Train | #Valid | #Test | #Snapshot |
|---|---|---|---|---|---|---|
| ICEWS14 | 7,128 | 230 | 74,845 | 8,514 | 7,371 | 365 |
| ICEWS18 | 10,094 | 256 | 373,018 | 45,995 | 49,545 | 365 |
| ICEWS05-15 | 23,033 | 251 | 368,868 | 46,302 | 46,159 | 4,017 |
| GDELT | 7,691 | 240 | 1,734,399 | 238,765 | 305,241 | 2,975 |

### 5.1 DATASETS

We evaluate our framework on four widely used temporal knowledge graph benchmarks: ICEWS14 (Garcia-Duran et al., 2018), ICEWS18 (Jin et al., 2019), ICEWS05-15 (Li et al., 2021), and GDELT (Leetaru & Schrodt, 2013). The ICEWS datasets consist of large-scale international political events, while GDELT is constructed from global news media. Following standard practice, we adopt a chronological $8:1:1$ split for training, validation, and testing, ensuring no temporal leakage (i.e., validation follows training, and testing follows validation). Summary statistics are reported in Table 1, and detailed descriptions are provided in the appendix.

**Baselines** To demonstrate the effectiveness of DSEE-MDiff, we compare against a broad set of recent TKG extrapolation models, covering recurrent, graph-based, rule-based, and diffusion-based approaches. The baselines include RE-NET (Jin et al., 2020), CyGNet (Zhu et al., 2021), xERTE (Han et al., 2020a), TITer (Sun et al., 2021), RE-GCN (Li et al., 2021), CEN (Li et al., 2022b), TiRGN (Li et al., 2022a), HiSMatch (Li et al., 2022c), RETIA (Liu et al., 2023a), CENET (Xu et al., 2023b), ICL (Lee et al., 2023), PPT (Xu et al., 2023a), LogCL (Chen et al., 2024), Re-Temp (Wang et al., 2023), DiffuTKG (Cai et al., 2024), CognTKE (Chen et al., 2025), and DSEP (Deng et al., 2025).

Table 2: The prediction performance (%) of MRR and Hits@1, 3, 10 on ICEWS14, ICEWS18, ICEWS05-15 and GDELT with time-aware metrics. The best results are highlighted in **bold** and the second-best results are underlined.

| Model | ICEWS14 | | | | ICEWS18 | | | | ICEWS05-15 | | | | GDELT | | | |
|---|---|---|---|---|---|---|---|---|---|---|---|---|---|---|---|---|
| | MRR | Hits@1 | Hits@3 | Hits@10 | MRR | Hits@1 | Hits@3 | Hits@10 | MRR | Hits@1 | Hits@3 | Hits@10 | MRR | Hits@1 | Hits@3 | Hits@10 |
| RE-NET (2020) | 36.93 | 26.83 | 39.51 | 54.78 | 28.81 | 19.05 | 32.44 | 47.51 | 43.32 | 33.43 | 47.77 | 63.06 | 19.62 | 12.42 | 21.00 | 34.01 |
| CyGNet (2020) | 35.05 | 25.73 | 39.01 | 53.55 | 24.93 | 15.90 | 28.28 | 42.61 | 36.81 | 26.61 | 41.63 | 56.22 | 18.48 | 11.52 | 19.57 | 31.98 |
| xERTE (2021) | 40.02 | 32.06 | 44.63 | 56.17 | 29.98 | 22.05 | 33.46 | 44.83 | 46.62 | 37.84 | 52.31 | 63.92 | 18.09 | 12.30 | 20.06 | 30.34 |
| TITer (2021) | 40.87 | 32.28 | 45.45 | 57.10 | 29.98 | 22.05 | 33.46 | 44.83 | 47.69 | 37.95 | 52.92 | 65.81 | 15.46 | 10.98 | 15.61 | 24.31 |
| RE-GCN (2021) | 40.39 | 30.66 | 44.96 | 59.21 | 30.58 | 21.01 | 34.34 | 48.75 | 48.03 | 37.33 | 53.85 | 68.27 | 19.64 | 12.42 | 20.90 | 33.69 |
| CEN (2022) | 42.20 | 32.08 | 47.46 | 61.31 | 31.50 | 21.70 | 35.44 | 50.59 | 46.84 | 36.38 | 52.45 | 67.01 | 20.39 | 12.96 | 21.77 | 34.97 |
| TiRGN (2022) | 44.04 | 33.83 | 48.95 | 63.84 | 33.66 | 23.19 | 37.99 | 54.22 | 50.04 | 39.25 | 56.13 | 70.71 | 21.67 | 13.63 | 23.27 | 37.60 |
| HisMatch (2022) | 46.42 | 35.91 | 51.63 | 66.84 | 33.99 | 23.91 | 37.90 | 53.94 | 52.85 | 42.01 | 59.05 | 73.28 | 22.01 | 14.45 | 23.80 | 36.61 |
| RETIA (2023) | 42.76 | 32.28 | 47.77 | 62.75 | 32.43 | 22.23 | 36.48 | 52.94 | 47.26 | 36.64 | 52.90 | 67.76 | 20.12 | 12.76 | 21.45 | 34.49 |
| CENET (2023) | 39.02 | 29.62 | 43.23 | 57.49 | 27.85 | 18.15 | 31.63 | 46.98 | 41.95 | 32.17 | 46.93 | 60.43 | 20.23 | 12.69 | 21.70 | 34.92 |
| ICL (2023) | – | 32.40 | 46.00 | 56.50 | 33.81 | 23.24 | 38.22 | 54.45 | 50.68 | 39.78 | 56.85 | 71.26 | 22.02 | 13.88 | 23.74 | 38.10 |
| PPT (2023) | 38.42 | 28.94 | 42.50 | 57.01 | 26.63 | 16.94 | 30.64 | 45.43 | 38.85 | 28.57 | 43.35 | 58.63 | – | – | – | – |
| LogCL (2023) | 48.87 | 37.76 | 54.71 | 70.26 | 35.67 | 24.53 | 40.32 | 57.74 | 57.04 | 46.07 | 63.72 | 77.87 | 23.75 | 14.64 | 25.60 | 42.33 |
| Re-Temp (2023) | 48.04 | 37.32 | 53.60 | 68.90 | 35.82 | 25.02 | 40.36 | 57.30 | 56.30 | 45.49 | 62.80 | 77.17 | 25.05 | 15.70 | **27.14** | **44.16** |
| DiffuTKG(2024) | - | - | - | - | 36.72 | 25.73 | - | 57.81 | 52.69 | 40.35 | - | 75.97 | **25.08** | **16.25** | - | 42.34 |
| CognTKE (2025) | 46.06 | 36.49 | 51.11 | 64.49 | 35.24 | 25.21 | 39.93 | 54.71 | 53.13 | 42.62 | 59.42 | 72.70 | - | - | - | - |
| DSEP (2025) | 44.87 | 34.35 | 50.29 | 64.86 | 33.81 | 23.24 | 38.22 | 54.45 | 50.68 | 39.78 | 56.85 | 71.26 | 22.02 | 13.88 | 23.74 | 38.10 |
| **DSEE-MDiff** | **51.49** | **40.28** | **57.82** | **73.18** | **37.73** | **26.30** | **42.66** | **60.49** | **58.07** | **47.05** | **65.06** | **78.74** | 24.48 | 15.17 | 26.51 | 43.46 |

### 5.2 MAIN RESULTS

We evaluate DSEE-MDiff on four standard TKG benchmarks (ICEWS14, ICEWS18, ICEWS05-15, and GDELT), with results summarized in Table 2. On ICEWS14, LogCL is the strongest baseline, yet DSEE-MDiff achieves the largest improvements, raising MRR and Hits@1 by 5.36% and 6.67%, respectively. This demonstrates the effectiveness of the dynamic–static encoder in fusing global and local cues. On ICEWS18, DSEE-MDiff outperforms DiffuTKG by 2.75% in MRR and 2.22% in Hits@1, showing that the Mamba-driven diffusion better preserves long-range dependencies. On ICEWS05-15, DSEE-MDiff remains ahead, with gains of +1.81% MRR and +2.13% Hits@1 over LogCL. On GDELT, although slightly below DiffuTKG, it surpasses most other baselines and achieves strong Hits@3 and Hits@10. Overall, DSEE-MDiff is competitive across all datasets and sets new

state-of-the-art results on three benchmarks, validating the synergy between the dynamic–static selector and the Mamba-based diffusion module.

Table 3: Ablation results on ICEWS14, ICEWS18, and ICEWS05-15. "w/o" indicates the component is removed. DES denotes the Dynamic Entity Selector; SES denotes the Static Entity Selector; DSES denotes the Dynamic–Static Entity Selector.

| Model | ICEWS14 | | | | ICEWS18 | | | | ICEW05-15 | | | |
|---|---|---|---|---|---|---|---|---|---|---|---|---|
| | MRR | Hits@1 | Hits@3 | Hits@10 | MRR | Hits@1 | Hits@3 | Hits@10 | MRR | Hits@1 | Hits@3 | Hits@10 |
| DSEE-MDiff | **51.49** | **40.28** | **57.82** | **73.18** | **37.73** | **26.30** | **42.66** | **60.49** | **58.07** | **47.05** | **65.06** | **78.74** |
| w/o DES | 33.82 | 26.06 | 38.44 | 47.68 | 23.29 | 16.72 | 26.72 | 35.78 | 36.21 | 27.90 | 40.95 | 51.79 |
| w/o SES | 38.26 | 28.25 | 42.47 | 58.19 | 29.05 | 19.53 | 32.52 | 47.71 | 39.13 | 29.13 | 43.55 | 58.87 |
| w/o DSES | 32.51 | 26.37 | 36.82 | 42.84 | 22.21 | 16.74 | 26.34 | 31.30 | 33.64 | 26.53 | 38.90 | 45.18 |
| w/o Mamba-Diffu | 49.59 | 38.37 | 55.58 | 71.15 | 36.61 | 25.22 | 41.41 | 59.45 | 56.79 | 46.08 | 63.51 | 76.89 |
| w/o Mamba | 51.24 | 39.88 | 57.65 | 73.07 | 37.34 | 25.89 | 42.25 | 60.30 | 57.70 | 46.82 | 64.62 | 77.83 |
| w/o Transformer | 50.37 | 38.90 | 56.43 | 73.05 | 37.35 | 25.93 | 42.18 | 60.19 | 57.67 | 46.70 | 64.64 | 78.12 |
| w/o Decoder | 47.23 | 35.66 | 53.17 | 70.07 | 34.69 | 23.49 | 39.23 | 57.21 | 55.40 | 43.87 | 62.51 | 77.41 |

## 5.3 ABLATION

To evaluate the contribution of each component in DSEE-MDiff, we perform ablations on ICEWS14, ICEWS18, and ICEWS05-15 (Table 3; "w/o" denotes removal). Removing the dynamic selector leads to a larger performance drop than removing the static selector, highlighting the dynamic branch as crucial for capturing global dependencies and temporal evolution, while the static branch provides complementary snapshot-level cues. Eliminating both selectors results in a greater decline than removing the Mamba-Diffusion module alone, showing that the selector pair plays the primary role in feature extraction and history aggregation. Within the Mamba-Diffusion module, removing either the Mamba or Transformer path produces comparable declines, suggesting their complementary roles in long-sequence modeling and context recovery. Finally, removing the decoder reduces performance across all datasets, confirming the importance of ConvTransE for temporal knowledge graph reasoning.

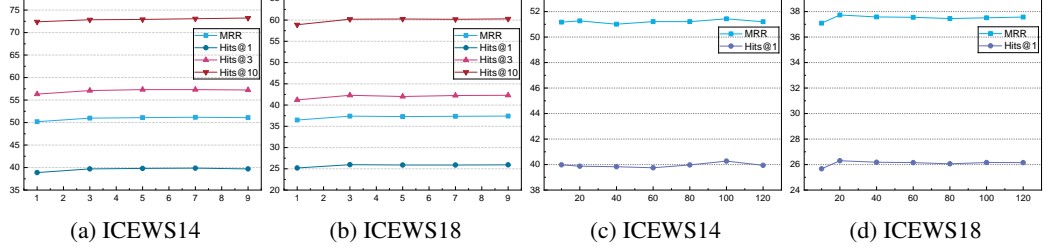

| (a) ICEWS14 | (b) ICEWS18 | (c) ICEWS14 | (d) ICEWS18 |
|---|---|---|---|

Figure 3: Effects of (a,b) historical window length and (c,d) fact sequence length on DSEE-MDiff performance on ICEWS14 and ICEWS18.

## 5.4 EFFECT OF HYPERPARAMETERS

To analyze the impact of hyperparameters, we vary (i) the length of the historical window and (ii) the input sequence length for the Mamba-Diffusion module on ICEWS14 and ICEWS18 (Fig. 3). On ICEWS14, performance converges at $l = 3$, and extending the window yields only marginal improvements, indicating that recent context is sufficient for accurate prediction. For the diffusion sequence length, ICEWS14 shows little benefit from longer sequences, whereas ICEWS18 exhibits clear gains when increasing from 10 to 20 steps before performance plateaus. We attribute this to a trade-off between context and noise: longer sequences introduce more query-irrelevant events, which increases noise and reduces denoising stability. Moderate sequence lengths therefore provide the best balance between information and robustness.

## 5.5 PERFORMANCE ON UNSEEN FACTS

To evaluate the ability of DSEE-MDiff to handle uncertainty from rare or previously unseen facts, we construct test splits on ICEWS14 and ICEWS18 that contain only facts absent from the training data.

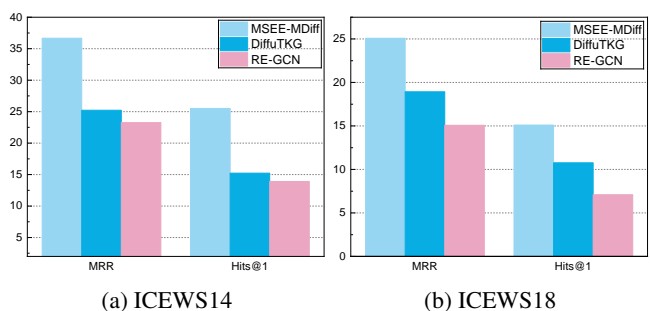

(a) ICEWS14        (b) ICEWS18

Figure 4: Performance on unseen facts on ICEWS14 and ICEWS18.

We compare against two representative baselines: DiffuTKG, which relies solely on diffusion, and RE-GCN, which emphasizes historical pattern modeling (Fig. 4). Across both datasets, DSEE-MDiff consistently achieves higher MRR and Hits@1, demonstrating its effectiveness in capturing semantic evolution from historical snapshots and leveraging it for extrapolation. Notably, under the higher uncertainty of sparse settings, DSEE-MDiff shows greater robustness, striking a better balance between exploiting informative context and mitigating noise.

Table 4: Case study on ICEWS14: Top-5 predictions for two queries (DSEE-MDiff and DiffuTKG).

| Query | Top-5 (DSEE-MDiff) | Top-5 (DiffuTKG) |
|---|---|---|
| s: Malaysia
r: Express intent to cooperate
o: ?
t: 334 | **Thailand**
China
Malaysia
Barack Obama
Iran | China
**Thailand**
Vietnam
Malaysia
Japan |
| s: Barack Obama
r: Make statement
o: ?
t: 353 | **North Korea**
Abdel Fattah Al-Sisi
Iraq
Iran
Government (Venezuela) | Iran
Japan
Iraq
Afghanistan
Benjamin Netanyahu |

## 5.6 CASE STUDY

To further examine the reasoning behavior of DSEE-MDiff, we select two real queries from ICEWS14 and compare the Top-5 predicted objects with DiffuTKG. Results are shown in Table 4, with the ground truth highlighted in bold.

In the first example, DSEE-MDiff ranks **Thailand** at #1, while DiffuTKG also hits the correct answer but places it at #2. This indicates that a diffusion-only pipeline is more prone to frequency bias and background noise when countries and interaction patterns are similar. By contrast, the dynamic–static entity selector enables DSEE-MDiff to capture structural and temporal cues more effectively, yielding a more stable ranking.

In the second example, DSEE-MDiff ranks **North Korea** at #1, whereas DiffuTKG fails to include the correct answer in its Top-5 list. This gap suggests that under sparse historical evidence and subtler relational semantics, DSEE-MDiff—through the joint modeling of long-range structure and local context in the dynamic–static selector, coupled with Mamba-driven diffusion that strengthens sequence dependencies—more accurately identifies time-relevant target entities.

## 6 CONCLUSION

We propose DSEE-MDiff to tackle the underuse of global latent semantics and the weak modeling of long fact sequences in TKG extrapolation. The framework integrates a dynamic–static entity selector, which fuses global semantics with local graph structure, and a Mamba-driven diffusion

module, which injects and removes noise to capture sparse and long-range dependencies more effectively. Their outputs are combined with a ConvTransE decoder for prediction and optimized jointly. Experiments on four public datasets show consistent improvements across multiple metrics, validating the effectiveness of the proposed approach.

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

## A APPENDIX

**Datasets** We conduct experiments on four real-world temporal knowledge graph datasets: ICEWS14 (Garcia-Duran et al., 2018), ICEWS18 (Jin et al., 2019), ICEWS05-15 (Li et al., 2021), and GDELT (Leetaru & Schrodt, 2013). The three ICEWS collections (ICEWS14, ICEWS18, ICEWS05-15) originate from the Integrated Crisis Early Warning System (ICEWS) and cover large-scale international political events spanning different time periods, whereas GDELT is compiled from global news media and records a broad spectrum of societal events across countries and regions. For all datasets, we adopt the standard chronological split protocol with an 8:1:1 ratio for train/validation/test (Zhu et al., 2021; Jin et al., 2019), strictly ensuring that all validation timestamps follow the training horizon and all test timestamps follow the validation horizon so as to prevent temporal leakage. Detailed dataset statistics, including the numbers of entities, relations, timestamps, and quadruples for each split, are reported in Table 1.

**Evaluation Metrics** We evaluate models using Mean Reciprocal Rank (MRR) and Hits@N, two widely used metrics for TKG reasoning. MRR measures the average inverse rank of the correct entity over all queries, placing greater weight on correctly ranked top candidates, while Hits@N reports the proportion of test queries for which the correct entity appears within the top-$N$ predictions. Higher values of both metrics indicate better performance. To enable fair and reproducible comparisons with prior work, we follow the same evaluation protocol as in the literature, including the standard filtered ranking setting and time-aware splits used by recent studies.

**Implementation Details** Unless otherwise stated, we use the Adam optimizer with a learning rate of 0.001 and set the embedding dimension to 200. The mini-batch size is adjusted dynamically according to the number of quadruples occurring at each timestamp to ensure stable training across datasets with different temporal densities. The optimal history window lengths on ICEWS14, ICEWS18, ICEWS05-15, and GDELT are set to 4, 5, 10, and 7, respectively. Within the diffusion module, the input sequence length is fixed to 20, 64, and 64 for ICEWS14/ICEWS18/ICEWS05-15, and to 20 for GDELT to accommodate dataset scale and sparsity. For the Mamba in the dynamic entity selection encoder, we set *channels* $= 128$, *state* $= 12$, *rank* $= 12$, and *kernel size* $= 2$. This configuration balances model capacity, speed, and memory usage. All experiments are implemented in PyTorch and executed on a workstation equipped with NVIDIA Tesla A100 GPUs (80,GB), ensuring consistent hardware and software environments across runs.

## A.1 MAMBA MODELS FOR DISCRETE DATA

SSMs are the core of Mamba. They map a multi-dimensional input sequence to an output sequence via latent states, parameterized by $(\mathbf{A}, \mathbf{B}, \mathbf{C}, \mathbf{D})$, which govern how inputs and current states determine the next state and output. A classical SSM is specified by the *state* and *observation* equations. Given input $\mathbf{x}_t \in \mathbb{R}$, output $\mathbf{y}_t \in \mathbb{R}$, and an $N$-dimensional hidden state $\mathbf{h}_t \in \mathbb{R}^N$,

$$\begin{aligned}
\mathbf{h}'_t &= \mathbf{A}\,\mathbf{h}_t + \mathbf{B}\,\mathbf{x}_t, \\
\mathbf{y}_t &= \mathbf{C}\,\mathbf{h}_t,
\end{aligned} \tag{22}$$

where $\mathbf{h}'_t$ is the time derivative of $\mathbf{h}_t$, and $\mathbf{A}, \mathbf{B}, \mathbf{C}$ are learnable matrices (a direct-feedthrough term $\mathbf{D}$ is optional).

For practical machine learning, continuous-time SSMs are discretized by converting continuous parameters to discrete ones. With step size $\Delta$, the continuous parameters $(\Delta, \mathbf{A}, \mathbf{B})$ can be mapped to discrete parameters $(\bar{\mathbf{A}}, \bar{\mathbf{B}})$ as

$$\begin{aligned}
\bar{\mathbf{A}} &= \exp(\Delta\mathbf{A}), \\
\bar{\mathbf{B}} &= \left(\Delta\mathbf{A}^{-1}\right)\left(\exp(\Delta\mathbf{A}) - \mathbf{I}\right)\Delta\mathbf{B},
\end{aligned} \tag{23}$$

where $\mathbf{I}$ is the identity matrix. This discretization enables efficient training and inference on discrete data while preserving the dynamical structure encoded by $\mathbf{A}$ and $\mathbf{B}$.

