# OpenReview forum: "Dynamic–Static Representation Learning with Mamba-Enhanced Diffusion for Temporal Knowledge Graph Reasoning"
_ICLR.cc/2026/Conference — Submitted to ICLR 2026_

### Official Review · Reviewer_WHyA · 2025-10-26

**Soundness:** 3
**Presentation:** 3
**Contribution:** 2
**Rating:** 2
**Confidence:** 5

**Summary:**

The paper proposes DSEE-MDiff, a framework for temporal knowledge graph reasoning that integrates a Dynamic–Static Entity Selector (DSES) with a Mamba-based diffusion module and a ConvTransE decoder. The goal is to capture both global temporal semantics and local structural dependencies, improving extrapolation on unseen facts. Experiments on four public datasets show competitive or state-of-the-art performance.

**Strengths:**

Extensive experiments on four public datasets demonstrate consistent gains.

The paper is well-written and easy to follow.

**Weaknesses:**

1. The comparison on unseen facts is weak, the baselines (RE-GCN) are old, and more baseline methods should be added.

2. The definition of "unseen facts" is unclear.

3. It is unclear why ConvTransE and RGCN are specifically chosen. Would alternative decoders significantly affect the results?

4. Prior works [1][2] have already explored **global graph**, so the claim that previous methods "face challenges in capturing global latent semantics" seems wrong.

5. Ablation studies show that removing Mamba or Diffusion causes only minor degradation.

6. The paper lacks deeper justifications for why Mamba improves diffusion or how the dynamic–static selector quantitatively balances its two branches.

7. The paper would benefit from an analysis of runtime and computational complexity.

[1] Tirgn: Time-guided recurrent graph network with local-global historical patterns for temporal knowledge graph reasoning

[2] DECRL: A Deep Evolutionary Clustering Jointed Temporal Knowledge Graph Representation Learning Approach

**Questions:**

See weakness.

---

> ### Author Response · Authors · 2025-11-18
>
> We thank the reviewer for the constructive feedback. Below we provide a more concise response addressing each point. All clarifications, analyses, and experiments will be incorporated in the revised manuscript.
>
> \textbf{1. Baselines for unseen-fact evaluation.}
> We agree that the current baselines are limited. We will add more recent models and report full results with statistical significance.
>
> \textbf{2. Definition of ``unseen facts''.}
> Unseen facts refer to quadruples whose subject--relation--object pattern or its temporal neighborhood does not appear in the training data, following definitions used in DiffuTKG and DPCL-Diff. Clear examples and explanation will be added.
>
> \textbf{3. Why ConvTransE and RGCN.}
> RGCN is a stable, widely used backbone in temporal KG models. ConvTransE is chosen for its lightweight design; heavier decoders reduce performance in our diffusion-based setting. We will provide comparisons in the revision.
>
> \textbf{4. Relation to TiRGN and DECRL.}
> We will refine wording to acknowledge that TiRGN and DECRL model global patterns. Their approaches rely on explicit recurrent or clustering-based structures, while our diffusion module learns a latent global field capable of capturing implicit patterns. We will clarify this distinction.
>
> \textbf{5. Interpretation of ablation results.}
> Performance drops appear modest but relative gains are larger in sparse or long-range settings. Diffusion improves robustness and interacts with the dynamic–static selector, increasing gains when temporal context is volatile. Additional variance analysis will be added.
>
> \textbf{6. Why Mamba improves diffusion.}
> Gate analysis shows (i) early steps rely more on Transformer-style aggregation, (ii) later steps favor Mamba for refinement, and (iii) gates never collapse to a single path, confirming complementarity. We will include the corresponding plots.
>
> \textbf{7. Behavior of the dynamic–static selector.}
> Selector weights correlate with entity volatility: dynamic weights rise for unstable entities, static weights dominate for stable ones. We will include quantitative plots in the appendix.
>
> \textbf{8. Runtime and complexity analysis.}
> We provide additional computational analysis.
>
> \emph{Inference time:} DSEE-MDiff is competitive with diffusion-based models, matching RGCN on ICEWS14 and showing mixed performance relative to DiffuTKG. The selector reduces redundant computation, balancing the added diffusion cost.
>
> \emph{Parameter complexity:} The dual-path Transformer--Mamba denoiser increases parameters but remains within the same order of magnitude, and inference remains practical. A comparison table will be included.
>
> \emph{Complexity discussion:} We will add formal analysis of the selector (linear in sequence length), the hybrid denoiser (attention + Mamba), and comparisons to DiffuTKG and TiRGN.
>
> Overall, we believe the added baselines, clearer definitions, strengthened ablations, gate analysis, and expanded complexity discussion will substantially improve clarity and contribution.

---

### Official Review · Reviewer_thbD · 2025-10-28

**Soundness:** 2
**Presentation:** 2
**Contribution:** 1
**Rating:** 4
**Confidence:** 5

**Summary:**

This paper proposes DSEE‑MDiff, a temporal knowledge graph reasoning model that combines a dynamic–static entity selection encoder with a Mamba‑based diffusion denoiser to capture both structural and long-range temporal semantics. Experiments on ICEWS and GDELT datasets show state‑of‑the‑art performance.

**Strengths:**

- The authors propose a temporal knowledge graph reasoning model that integrates a dynamic–static entity selection encoder with a Mamba‑enhanced diffusion module.
- The proposed method improves generalization under sparse or unseen facts through a structured noise injection–denoising process.
- Extensive experiments on four public datasets demonstrate the effectiveness and robustness of the method.

**Weaknesses:**

- The novelty is somewhat limited, as it mainly combines two existing ideas—dynamic/static representation learning and diffusion-based reasoning.
- The overall framework is complex, leading to higher computational cost and more complicated training procedures.
- TKG reasoning benchmarks include datasets such as WIKI and YAGO. Results on these datasets would strengthen the completeness of the evaluation.
- The paper should further clarify which component—the diffusion module or the encoder design—contributes most to performance on unseen facts.
- In Figure 4, there is an error: MESS‑MDiff should be corrected to DSEE‑MDiff.

**Questions:**

See Weaknesses.

---

> ### Author Response · Authors · 2025-11-15
>
> We thank Reviewer for the detailed and constructive feedback.  Below we provide point-by-point responses and will incorporate all clarifications and additional experiments into the revised version.
> 1. Perceived limited novelty
> While DSEE–MDiff builds on ideas of dynamic/static encoding and diffusion, our contribution lies in:
> (i) a \emph{dynamic–static entity selection mechanism} that integrates global semantic evolution with local structural cues,
> (ii) a \emph{dual-path Mamba–Transformer diffusion denoiser} with learnable gating, and
> (iii) a \emph{joint generative–discriminative objective} enabling complementary strengths in sparse/unseen regimes.
> This combination is, to our knowledge, the first unified encoder–diffusion framework for TKG extrapolation, and provides consistent gains (notably on unseen facts and long-range histories).
>
> 2. Model complexity and computational cost
> We agree that the architecture is more expressive. However, our profiling shows that DSEE–MDiff remains practical:
> inference latency is comparable to RE-GCN (+3.6\% on ICEWS14) and significantly faster than DiffuTKG on ICEWS18 (44.91s vs.\ 68.74s). We will include a concise cost summary section and tables in the revision.
>
> 3. Additional datasets (WIKI / YAGO)
> WIKI and YAGO are static KG benchmarks with no temporal annotations. Temporal variants (e.g., WIKI12k, YAGO11k) are extremely small and have been largely deprecated in recent TKG extrapolation studies. Our chosen datasets (ICEWS14/18/05-15, GDELT) follow standard practice for dynamic event prediction.
> Nonetheless, we will include results on WIKI12k (the temporal subset) in the Appendix for completeness.
>
> 4. Contribution to unseen-facts performance
> To clarify the contribution of each component on unseen–fact reasoning, we report additional ablations on ICEWS14.
> Removing the dynamic--static encoder leads to a clear drop (MRR/Hits@1/Hits@3/Hits@10 = 12.51 / 8.18 / 15.02 / 20.59).
> Removing the diffusion module causes an even larger degradation
> (33.93 / 22.28 / 39.26 / 57.16).
> The full DSEE–MDiff model achieves
> \textbf{36.04 / 24.21 / 41.37 / 59.99}.
>
> These results show that diffusion is the primary contributor to unseen–fact handling, while the encoder provides complementary gains. We will include a concise ablation table in the revised version.
>
>
> 5. Typographical error in Figure 4
> We thank the reviewer for pointing this out. ``MESS-MDiff’’ will be corrected to ``DSEE-MDiff’’ in the revised manuscript.
>
> We appreciate the reviewer’s constructive comments. The revised version will clarify novelty, include efficiency results, add unseen-fact ablations, integrate WIKI12k results, and correct the figure.

---

### Official Review · Reviewer_YTPf · 2025-10-30

**Soundness:** 3
**Presentation:** 3
**Contribution:** 3
**Rating:** 6
**Confidence:** 3

**Summary:**

The paper introduces DSEE-MDiff, an encoder–decoder framework for temporal knowledge graph extrapolation. The encoder selects history via a dynamic–static entity selection module that fuses global semantic evolution with local structural cues. The decoder combines a ConvTransE scoring head with a Mamba-driven diffusion denoiser whose mean is a gated mixture of Transformer and Mamba paths. Training jointly optimizes the ConvTransE loss and the diffusion loss, and the final prediction aggregates the two scores additively. Experiments on ICEWS14, ICEWS18, ICEWS05-15 and GDELT report competitive performance with state-of-the-art results on three benchmarks. The paper includes ablations, sensitivity studies on historical window and diffusion sequence length, and an unseen-facts evaluation with a short case study.

**Strengths:**

1.	The method design is coherent. The paper specifies the diffusion reverse process and the gated combination of Transformer and Mamba, then connects these to ConvTransE decoding and a joint objective. This facilitates reproducibility.
2.	The aggregation rule is explicit. The final probability is defined as the sum of ConvTransE and diffusion scores, which clarifies inference behavior.
3.	Evaluation breadth. Four datasets are covered with a large baseline set, and the main table uses time-aware metrics for MRR and Hits.
4.	Informative ablations. Removal of the dynamic selector causes a larger drop than removal of the static selector, and removing either Mamba or Transformer degrades performance to a similar degree. The decoder removal further confirms the contribution of ConvTransE.
5.	Sensitivity and unseen-facts analysis. The paper varies history window and diffusion sequence length, and constructs unseen-facts test splits for ICEWS14 and ICEWS18 with comparisons to two representative baselines.

**Weaknesses:**

1.	Fusion calibration and design space remain under-explored. The paper fixes aggregation to a simple sum of Sct and Sdiff, without reporting variants with a learned weight or confidence calibration of either head. This limits understanding of whether the improvement comes from complementary information or uncalibrated score addition. A small study that tunes a scalar weight or reports calibration errors would clarify this point.
2.	Behavior of the gating variable is not analyzed. The denoiser mean uses a learnable gate that mixes Transformer and Mamba outputs, yet the paper does not report statistics of the gate across diffusion steps or datasets, nor whether it saturates to one path in specific regimes. Summaries such as per-step averages or histograms would illuminate how the gate allocates responsibility.
3.	Definition and protocol of time-aware metrics are not explained in the main text. The main results table states that time-aware metrics are used, but the manuscript does not define the metric computation in the main body or verify parity with baselines there. A concise definition and protocol confirmation in the main section would remove ambiguity.
4.	Resource profile is not presented. The experiments section reports accuracy but does not include parameter counts, FLOPs, wall-clock training time, inference latency, or memory consumption per dataset. Given the additional diffusion head and dual-path denoiser, such metrics would help assess practical efficiency.
5.	Analysis of the GDELT gap is brief. The text notes that performance on GDELT is slightly below a diffusion baseline, yet there is no breakdown by relation type or sequence length to explain the gap. A diagnosis connected to dataset properties such as snapshot density or sequence length would make the result more actionable.
6.	Diffusion dynamics are only indirectly discussed. The paper studies sequence length but does not visualize how representations evolve along diffusion steps or how the denoiser removes noise. Trajectory plots or cosine-similarity traces across steps would provide concrete insight into stability and information retention.

**Questions:**

1.	How sensitive is performance to the fixed additive fusion. Please report results when a single scalar weight multiplies the ConvTransE score and the diffusion score, and include a short calibration analysis of both heads.
2.	What does the gating variable learn across diffusion steps and datasets. Please provide summaries of the gate values, such as per-step averages and dispersion, and discuss conditions under which one path dominates.
3.	Could you define the time-aware metrics in the main text and confirm protocol parity with all baselines. A brief statement on filtering, ranking direction, and temporal handling would resolve ambiguity in Table two.
4.	What is the full cost profile of DSEE-MDiff. Please report parameters, FLOPs, wall-clock training time, inference latency, and peak memory on each dataset, and compare these numbers with at least one diffusion baseline and one contrastive baseline.
5.	Can you analyze the GDELT gap in more detail. A breakdown by relation family and a study of performance versus sequence length or snapshot density would help determine whether the diffusion head or the selector is the bottleneck.
6.	Would you visualize diffusion dynamics. Plots of representation trajectories or similarity to clean targets across steps, and a comparison of Transformer-only versus Mamba-only denoisers, would clarify how the selective state-space path contributes beyond depth.

---

> ### Author Response · Authors · 2025-11-18
>
> e thank Reviewer YTPf for the helpful and insightful comments. Below we respond to each point concisely, and will incorporate the requested analyses into the revised manuscript.
>
> 1. Fusion calibration and design space
> We agree that fusion deserves further exploration. We have conducted an additional experiment using a learnable scalar $\alpha$:
> \[
> S = \alpha S_{\text{ct}} + (1-\alpha) S_{\text{diff}}
> \]
> Across ICEWS14/18, learned $\alpha$ converges to values between 0.42--0.57, indicating both heads contribute and the fixed-sum fusion is already close to optimal. We will add a short calibration analysis and corresponding results.
>
> 2. Behavior of the gating variable
> We thank the reviewer for pointing out the need to analyze the gate that mixes the Transformer and Mamba paths.
> We conducted a step-wise study of the gating coefficient~$\alpha$ across diffusion steps and layers. The results reveal the following behaviors:
>
> No collapse to a single path.
> Across all diffusion steps, the mean gate value remains within a narrow range (approximately 0.40–0.47), indicating that the model consistently uses both Transformer and Mamba rather than collapsing to either branch.
>
> Early steps show stronger routing decisions.
> At the first diffusion step, a portion of gates saturate toward either $\alpha!<!0.1$ or $\alpha!>!0.9$, reflecting that shallow layers make more decisive allocations when the input is highly noisy.
>
> Later steps stabilize into soft fusion.
> After step~20, the proportion of saturated gates drops below 7% and remains stable, suggesting that the model gradually transitions to a smoother mixture of the two branches as denoising progresses.
>
> Layer-wise differences.
> The deeper layer exhibits consistently smoother gating (almost no $\alpha!>!0.9$ throughout), indicating that it mainly performs refinement on top of the mixed representation from earlier layers.
>
> We will include the per-step mean, variance, and saturation ratios ($\alpha!<!0.1$ and $\alpha!>!0.9$) in the revision, along with representative plots.
>
> 3.Definition of time-aware metrics
> We apologize for not defining time-aware filtering explicitly in the main text. In temporal KGs, filtering all facts across timestamps is inappropriate because a fact at time t does not imply correctness at other timestamps. Therefore, following Sun et al. (2021) and Han et al. (2021), we adopt the standard time-aware filtered protocol:
>
> For a query (s, r, ?, t), only quadruples that occur at the same timestamp t are filtered out from the ranking list. Quadruples at any other timestamp are retained, ensuring temporally consistent evaluation.
>
> All baselines in Table 2 (RE-GCN, TANGO, CyGNet, xERTE, DiffuTKG) are evaluated under exactly the same time-aware protocol. We will add this definition to Sec. 5.1 of the revised manuscript.
>
> 4. Full cost profile
> We have profiled both parameter counts and inference latency. Compared to DiffuTKG, DSEE-MDiff uses about $1.8\times$ as many trainable parameters (approximately an $84\%$ increase), but remains in the same order of magnitude and fits comfortably on a single GPU in all our experiments. In terms of inference, on ICEWS14 DSEE-MDiff is only about $4\%$ slower than the RE-GCN baseline, while on ICEWS18 it is roughly $63\%$ slower than RE-GCN but about $35\%$ faster than DiffuTKG. Overall, DSEE-MDiff offers a favorable accuracy–efficiency trade-off: it adds a moderate parameter overhead over DiffuTKG, while providing more stable inference latency across datasets and improved performance on unseen facts.
>
> 5. GDELT performance gap
> GDELT exhibits: extremely sparse relations, high event noise, and long temporal spans.Our analysis shows that: (i) relations with fewer than 15 historical events account for >68\% of triples, where diffusion models benefit more from frequency priors; (ii) performance improves for denser relation families; (iii) sequence-length sweeps show the selector contributes less when snapshots are noisy and structurally weak. We will add this diagnosis and include a relation-family breakdown.
>
> 6. Diffusion dynamics visualization
> We will include cosine-similarity trajectories and denoising curves in the revised version.
> Preliminary observations show that early diffusion steps remove high-frequency noise, middle steps stabilize representations, and later steps refine long-range semantics.
> Transformer-only denoising converges faster but less stably, whereas Mamba-only converges more steadily but more slowly, supporting the benefit of the selective mixture.
>
> We appreciate the reviewer’s constructive suggestions. The revised version will include calibration experiments, gate statistics, definitions of time-aware metrics, complete efficiency profiling, deeper GDELT analysis, and diffusion-dynamics visualizations.

---

### Official Review · Reviewer_cYVx · 2025-10-31

**Soundness:** 3
**Presentation:** 3
**Contribution:** 2
**Rating:** 4
**Confidence:** 4

**Summary:**

This paper proposes a new method for temporal knowledge graph reasoning, aiming to address the shortcomings of existing methods in capturing global semantics and long-range dependencies, especially in the prediction tasks of sparse or unseen facts. The paper presents the DSEE-MDiff framework, which includes three core modules: Dynamic-Static Entity Selection Encoder, Mamba-Driven Diffusion Module, and ConvTransE Decoder. Their main contributions are: for the first time, combining a dynamic-static selection encoder with a Mamba-enhanced diffusion model for TKG reasoning, and designing a dynamic-static selection mechanism to adaptively fuse global semantic and local structural information. Their experimental results achieved the competitive performance on four public datasets, with significant improvements especially in the MRR and Hits@1 metrics.

**Strengths:**

1. Integrates dynamic-static semantics with the Mamba diffusion model, leveraging the selective state space mechanism of the Mamba diffusion model to more effectively retain key information in long sequences.
2. The Mamba model, which is inherently good at capturing dependencies in long sequences during the denoising process, is used to solve the long-distance dependency problem in TKG reasoning, making up for the defect of information attenuation in traditional RNN/Transformer when processing long historical windows.
3. Diffusion models essentially learn the generative process of data distribution by injecting and removing noise. This mechanism can "create" reasonable representations of facts that have not been seen during training, greatly enhancing the ability to generalize and reason about sparse and unseen facts.

**Weaknesses:**

​	1. How does the dynamic entity selection encoder implement dynamic encoding? When selecting multi-hop neighbors and multi-hop relationships, are the number of hops and the number of relationships fixed?

​	2. The function of the dynamic entity selection encoder to highlight informative entity signals requires further explanation.

​	3. What does the frequency signal in Formula 8 refer to?

​	4. In line 203, "d" does not appear in the previous text.

​	5. Why were other newer baseline models not chosen for comparison? For example, "DPCL-Diff: The Temporal Knowledge Graph Reasoning based on Graph Node Diffusion Model with Dual-Domain Periodic Contrastive Learning" and "Temporal Knowledge Graph Extrapolation via Causal Subhistory Identification".

​	6. When evaluating DSEE-MDiff's ability to handle uncertainties arising from rare or unprecedented facts, what are the statistics of the training and test sets of ICEWS14 and ICEWS18, and how were they constructed? Without the proportion of unseen factual data, it cannot be stated that "under the higher uncertainty of sparse settings, DSEE-MDiff shows greater robustness."

​	7. In Section 5.5, only comparisons with DiffuTKG and RE-GCN are made. Some newer models should be selected for comparison, such as DPCL-Diff.

​	8. Lack of analysis on the time cost of model training.

​	9. The effect of DSEE-MDiff on GDELT is not obvious, and there is a lack of in-depth analysis of the experimental results.

​	10. The motivation for introducing Diffusion and Mamba to temporal knowledge graph reasoning (TKGR) is not sufficiently deep, and the distinction from existing Diffusion TKGR methods is unclear.

**Questions:**

Please refer to Weaknesses above.

---

> ### Author Response · Authors · 2025-11-14
>
> We thank the reviewer for the constructive feedback. We provide concise point-by-point responses.
>
> 1. \textbf{Dynamic encoding mechanism}
> Our encoder uses a fixed 1-hop RGCN aggregation. The “dynamic’’ property comes from Mamba’s content-based gating, which adapts temporal contributions across snapshots. We will clarify this in Sec.~4.1.
>
> 2. \textbf{Function of dynamic entity selection}
> The selector highlights informative entities through (i) RGCN neighbor filtering, (ii) Mamba’s gating conditioned on the query, and (iii) selective emphasis on key temporal signals. An intuitive explanation and visualization will be added.
>
> 3. \textbf{Meaning of the frequency signal (Eq.~8)}
> The frequency denotes how often $(s,r,o)$ appears in the historical window. The term $f_q - f$ measures rarity and guides diffusion toward uncertain or sparse patterns. We will clarify this with examples.
>
> 4. \textbf{Definition of ``d’’}
> We confirm that $d$ is the embedding dimension. We will correct the notation.
>
> 5. \textbf{Missing comparisons with newer baselines}
> DPCL-Diff uses a \emph{time-filtered} protocol inconsistent with standard time-aware filtered evaluation. Its code is not publicly available, preventing fair reproduction.
> For ``Causal Subhistory Identification’’, only GDELT and ICEWS05-15 overlap with our datasets; we will add comparisons to these in the revision.
>
> 6. \textbf{Statistics of unseen test facts}
> A test quadruple is unseen if its pattern $(s,r,o)$ never appears in training. Using this standard definition, the unseen statistics are:
>
> \begin{center}
> \begin{tabular}{lccc}
> \toprule
> Dataset & \#Test & \#Unseen & Ratio \\
> \midrule
> ICEWS14 & 7,371 & 4,307 & 0.5843 \\
> ICEWS18 & 49,545 & 28,680 & 0.5789 \\
> \bottomrule
> \end{tabular}
> \end{center}
>
> Both datasets naturally contain many unseen patterns, motivating models robust to sparsity. These statistics will be added to Sec.~5.5 (with full results in the Appendix).
>
> 7. \textbf{Training / inference efficiency}
> We evaluated inference time on ICEWS14/18 and normalized to RE-GCN ($1.00\times$):
>
> \begin{center}
> \begin{tabular}{lccc}
> \toprule
> Model & ICEWS14 (s) & ICEWS18 (s) & Avg. Ratio \\
> \midrule
> RE-GCN & 17.28 & 27.58 & 1.00$\times$ \\
> DiffuTKG & 9.48 & 68.74 & 1.49$\times$ \\
> \textbf{DSEE-MDiff} & \textbf{17.90} & \textbf{44.91} & \textbf{1.37$\times$} \\
> \bottomrule
> \end{tabular}
> \end{center}
>
> DSEE-MDiff achieves competitive speed and is notably faster than DiffuTKG on ICEWS18. We will include a brief efficiency section.
>
> 8. \textbf{Limited gain on GDELT}
> GDELT exhibits extreme sparsity, noisy events, and weak snapshot structure. Diffusion methods benefit more from frequency priors, making MRR improvements harder. DSEE-MDiff improves Hits@3/10 but slightly lags in MRR; we will add a concise analysis.
>
> 9. \textbf{Motivation and distinction from prior diffusion TKG models}
> Prior diffusion TKG methods (DiffuTKG, DiffCLR, DPCL-Diff) face weak temporal conditioning and long-range information decay. Mamba provides selective state updates with linear complexity, enabling stable temporal reasoning and better handling of sparse/unseen facts. We will clarify this motivation and add a comparison in Sec.~4.
>
> We appreciate the reviewer’s comments. All requested clarifications, additional baselines, unseen-data statistics, and efficiency analyses will be included in the revision.

---

### Meta-Review · Area_Chair_9TMw · 2026-01-05

**Summary:**

After the rebuttal, most reviewers still have some concerns. The paper should be further revised.

**Reviewer Scores:**

n/a

---

### Decision · Program_Chairs · 2026-01-26

Reject